# High-resolution raindrop counting via instantaneous frequency sensing on hydrophobic elastic membranes

**Rytis Paškauskas** ⓘ *

Science, Technology and Innovation Unit, The Abdus Salam International Centre for Theoretical Physics (ICTP), Trieste, Italy

* rytis.paskauskas@ictp.it

## Abstract

In this paper, we introduce a novel approach that paves the way for the creation of affordable, high-precision rainfall sensors utilizing microphone data. The cornerstone of this methodology is an innovative algorithm capable of converting audio recordings into distinctive features, which are subsequently processed by a compact machine learning model. Our findings demonstrate that this technique can attain a temporal resolution of 10 milliseconds with an accuracy of 80%, underscoring its potential to overcome the limitations imposed by the necessity for power infrastructure and specialized expertise in traditional rain sensing methods.

**Data Availability Statement:** Data is available at https://github.com/rytis-paskauskas/RainDropCounter. The folder 'data' contains data used in the article. The folder 'sandbox' contains python code snippets to aid in reproducing the article's results.

## 1 Introduction

A disdrometer is a precision instrument that counts and measures drop sizes of natural rain [1, 2]. An acoustic disdrometer works by converting the mechanical energy of an impact on a surface to an equivalent electrical pulse. The well-known shortcomings of a *passive* measurement method were summarized already in a seminal work introducing the industry standard Joss-Waldvogel Disdrometer (JWD) [1]: *"The amplitude of the electrical pulses depends on where the raindrop falls on the membrane; The temporal resolution of approx. 30 ms is insufficient, . . .and splashes from large drops are counted as small drops (translated from the original German)"*. An important technological breakthrough of JWD improved the temporal resolution by using electromechanical feedback loop to compensate the force of impact [2]. However, the high cost, infrastructure requirements (such as an AC source), and the need for technical expertise can be barriers for its widespread use.

Simpler, less expensive, piezoelectric transducer-based rainfall measurement instruments rely on amplitude thresholding [3, 4]: a time window where the response exceeds a certain magnitude is 'locked' for calculations, during which the subsequent drops are either not counted (locked out) or incorrectly counted [2]. Linear relaxation broadens the acoustic response of a physical impact, limiting the temporal resolution. To reduce the lock-out window, materials with large attenuation coefficients are used (plastic [4], aluminium [5], stainless steel [6], glass [7]). Unfortunately this also reduces sensitivity and may lead to small droplet under-counting [2, 8, 9].

**Funding:** The author(s) received no specific funding for this work.

**Competing interests:** The authors have declared that no competing interests exist.

Although potentially more sensitive, elastic materials have not been considered due to their noisy acoustic response (the top pane of Fig 1 illustrates the case in point using a HDPE membrane), material degradation, high risk of damage, and limitation to liquid-only hydrometeors. On the other hand, devices with soft membranes are easy to make or replace with highly available materials. This practical advantage as well as the simplicity and versatility of microphone-based, contactless sensing that does not require specialized electronic circuitry would be of value where ultra low cost, short-term campaigns or educational and outreach goals are prioritized. Among the latter we consider TinyML education [10] and affordable weather station projects [11–13] as particularly relevant to our work. In this article, we describe such a device and its data analysis methodology.

Our main contribution is a new proposal to identify impacts with high frequency transients in membranes made with elastic hydrophobic materials. By relying exclusively on frequency detection this method is, in principle, tolerant to low frequency noise and is time limited only by the observed $\sim 5$ millisecond transient time. The short lock out window and the decoupling from low frequency spectrum are its main advantages over the amplitude thresholding method. To further emphasize this point the impact detection is implemented without performing amplitude calculations, achieving temporal resolution of about 10 milliseconds with the added benefit of high impact sensitivity. The sensing surface construction and the phenomenology are described in Sects. 2 and 3. Our contribution includes a concrete implementation theory using two original components: A lightweight *acoustic feature model* to detect instantaneous high frequency vibrations, as described in Sect. 4.4, and a machine learning

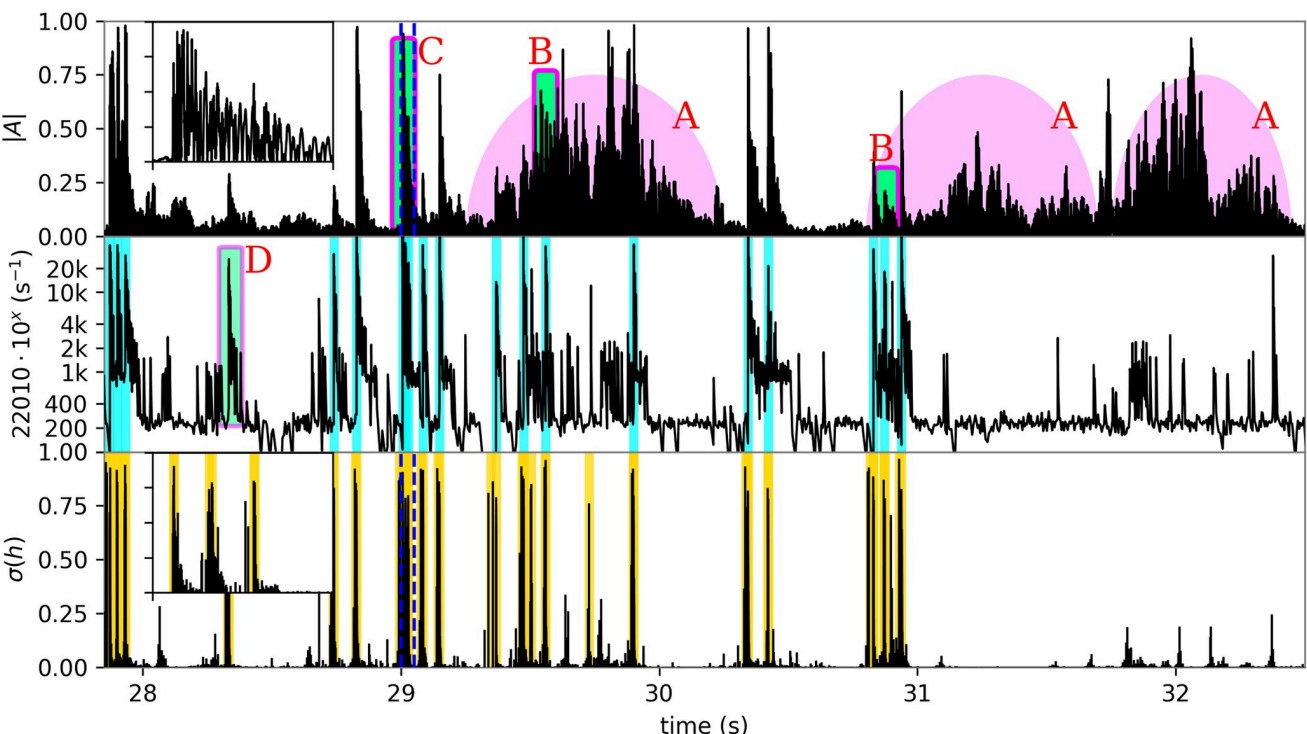

**Fig 1. Challenges of detection with a soft membrane and the proposed solution.** Top pane: a noisy amplitude profile that poses several challenges to amplitude thresholding. Wind (A) is a rich source of false positives and noise, concealing true events (B). Large lock out windows (C and inset) 'buries' some impacts (inset expands a 50 ms window containing 3 events). Middle pane: the acoustic feature *x* is good at discriminating high frequency transient from noise. Vertical bars (cyan): annotations, D: suspected missed annotation (label noise). Bottom panel: convolutional neural networks (CNN) are used to improve the predictions. Vertical bars (yellow): predictions by cluster mean method.

framework that uses compact, physics-informed *convolutional neural networks* (CNN) to improve the quality of detection, described in Sect. 4.5. Our implementation targets embedded ML ready devices [10] capable of 44.1kHz audio capture (a wide range of consumer microcontrollers and all smartphones fit this bill) and is intended to run 'near the source', on small and resource constrained microcontrollers. This is achieved by design considerations taking into account TFLite Micro framework operator support [14], small CNN models with $\mathcal{O}(10^2)$ parameters, and half floating point data precision. We provide background for the statistical model in Sect. 4.1 and propose a rigorous self-evaluation methodology and criteria in Sect. 4.2. Several CNN architectures are compared from the point of view of high frequency noise suppression, which remains a challenge.

Our long term goal is an ultra low cost TinyML enabled rainfall measurement device. The results so far and suggestions for moving forward are discussed in Sect. 5. The present methodology can be applied to count impact arrivals directly [15]. With the addition of mean drop size calibration, the rainfall rate estimation is feasible [16, 17]. An addition to the present methodology will be sought for the estimation of each droplet's kinetic energy. The last two steps, as well as field testing are outside the scope of the present work that focuses on describing a new detection principle. We believe the underlying causes of high frequency transients to be related to droplet-surface interaction as the former disintegrates upon impact. Despite significant recent advances [18–21] the setting of high velocity droplets impacting elastic hydrophobic surfaces, to our knowledge has not been studied from the point of view of rainfall metrology. A more detailed understanding of microphysical effects in this context would provide valuable insights into useful material parameter ranges, etc.

The interest to use microphones for rainfall monitoring is growing, with applications in indirect rainfall measurement [22–25], biodiversity monitoring [26–28] (where rainfall is a nuisance parameter) and leveraging existing surveillance data [29, 30]. A few studies applied deep learning such as convolutional [25, 31], recurrent [32], and attention based neural networks [29]. Our work addresses similar questions. The previous studies target rainfall classification using a few states of (e.g. no, light, moderate, heavy) rain from, typically, unstructured 'soundscape' recordings. Our methodology aims at counting each droplet on a purpose made enclosure but does not yet directly estimate the rainfall rate. The previous studies use acoustic deep learning methodologies with generic features (several are explored in [31]) and benefit from well studied feature models and deep learning architectures, but tend to result in memory and computational requirements that are not feasible for small microcontrollers. We use physics informed features and CNN models which may require additional studies, but have so far resulted in extremely frugal algorithms, with most of examined CNNs having less than 1000 weights.

## 2 Hardware and software

The hardware for an external casing is illustrated in Fig 2, pane 1. We use a standard 100 mm diameter PVC pipe, one end of which is covered with a sturdy plastic wrap to serve as a sensing membrane. We used metalized Mylar (polyethylene terephthalate; PET) and high-density polyethylene (HDPE), both of which are easily available. The advantages of PET (Young modulus $E \approx 2.0$ GPa, density $\rho \approx 1.35$ g/cm$^3$, thickness $d \approx 0.01$ mm) are good physical and chemical stability, humidity and heat protection. HDPE ($E \approx 0.5$–1 GPa, $\rho \approx 0.95$ g/cm$^3$, $d \approx 0.1$ mm) can be obtained from recycled materials, e.g. shopping bags, and be sufficient for short campaigns. The enclosing pipe is cut into segments that are long enough to house a smartphone. The wrap is fastened to the pipe using stationery and duct tapes. The goal is a smooth, well tempered surface with uniform stress. Taking into account 97mm internal

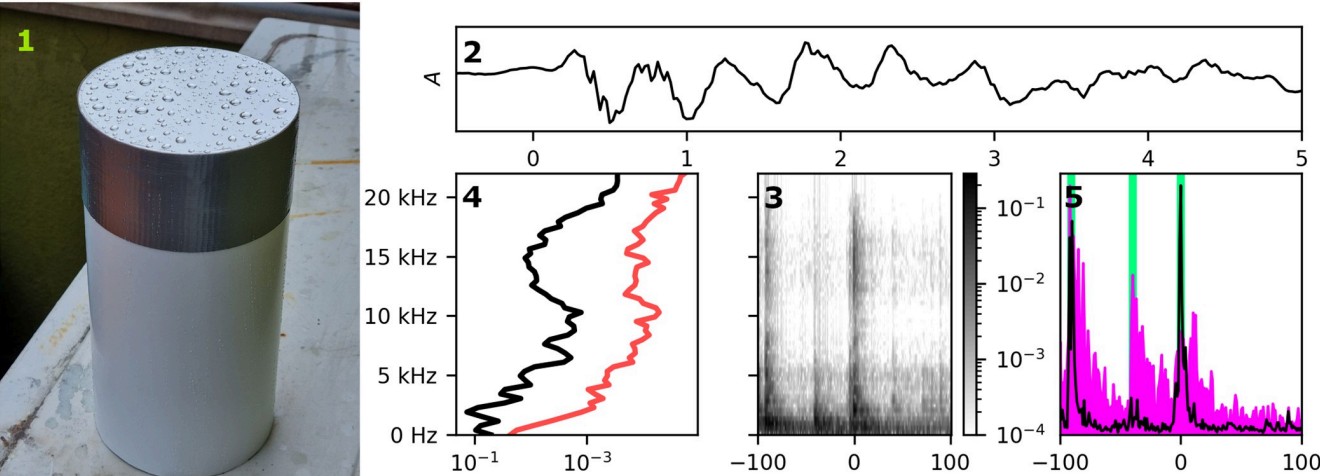

**Fig 2. External casing and acoustic signatures of a raindrop impacting its surface, 2: Time series of a representative impact with visible high frequency vibration.** 3: STFT logarithmic spectrogram, showing 100 ms on both sides of the impact. 4: Frequency filter banks, averaged over 5 steps (5 ms) centered on (black) and off (red) the impact. Note a 12–18 kHz mode on the impact. 5: Different contrasts averaged over high frequency (black) and low frequency (red) bands. Vertical bars (green): annotated events.

diameter, the sensing surface has an area of 73.9cm$^2$. The assembly, excluding the cutting of a pipe, is a simple and safe process that does not require technical equipment.

Data preparation and handling are conducted entirely within Python 3.11 [33]. The SoX [34] and Audacity [35] software were used for signal processing and visualization, respectively. The deep learning networks are developed with TensorFlow v2.15 [14]. The network training was done on NVIDIA GPUs. All other computing was performed on a CPU-only laptop running a Linux OS. Data, code samples and additional instructions are available at https://github.com/rytis-paskauskas/RainDropCounter.

## 3 Drop microphysics

The subsequent methodology hinges on an observation, made on several hydrophobic soft materials, that a high frequency vibration is concomitant with a physical impact. An example of the acoustic response is illustrated in Fig 2, panes 2–5. In pane 2, a tenuous transient is visible for about 1 ms, superimposed on a 'slow' relaxation of a membrane's mode. Pane 3 illustrates the same impact using short time Fourier transform (STFT) with 128-sample Hamming windows (2.67 ms) and 1 ms temporal steps. The reference (at zero) and several surrounding impacts are identifiable as narrow vertical lines across the spectrum. Pane 4 shows two slices of the spectrogram 'horizontally averaged' over 5 time steps: on impact (black line) and off impact (red line, +60 ms offset). All these observations suggest the existence of a broad mode in the 12–18 kHz range, which we found to characterize many impacts.

Lastly, two band averages (magenta: low frequency, black: 12–18 kHz band), shown in pane 5, suggest that the high frequency band displays a better contrast. In Sect. 4.4 we use this insight to develop a feature model for high frequency detection without the use of Fourier transforms.

The high-frequency transient was observed with both PET and HDPE materials, using pipes of various diameters and different surface tension levels across trials, due to the inability to precisely control surface tension in our DIY setup. These widely varying parameters influenced the resonance frequencies, which stayed within the 1 kHz range, but seemingly had no

effect on the high-frequency transient. To our understanding, the time scales and frequencies involved do not fit a conventional picture of impulse response of an elastic membrane. The presence of high frequencies suggest that scales comparable to the droplet size might be involved. Recently there has been considerable interest in soft surfaces [20, 21]. The most common natural rain drop diameters $D$ vary in the range 0.2–5.0 mm [36]. The terminal velocity is well approximated by $V(D) = 9.55(1 - \exp(-0.6D))$ [37]. Even though the high terminal velocity generally leads to splashing [19], additional effects might be in place due to the hydrophobic and elasticity factors. The microphysics of liquid droplet impacting surfaces has longstanding interest [18, 19]. In a different context it has been suggested, for example, that soft surfaces impacted by liquid droplets trap more air underneath than their rigid counterparts, increasing the possibility of surface-droplet interactions through capillary waves [21]. It is presently unknown whether a microphysical effect is the cause of the observed high frequency mode. Additional high resolution experiments could help to identify the optimal material properties. However, even without detailed understanding of the causes, we are able to leverage the apparent acoustic effect and devise a methodology that is based solely on local frequency detection. Its reliance on frequencies increases the robustness to low frequency amplitude noise, whereas the short duration of the transient improves the temporal resolution.

## 4 Methodology

Rain drop impacts are routinely counted by disdrometers, and databases of aggregated data are available [38]. Our setting has specific requirements: impacts should land on a soft hydrophobic surface (as described in Sect. 2) and high resolution data is required; The sampling frequency should be at least 44,100 Hz. Therefore we first commit a few recordings with annotated impacts to a dataset.

We begin the analysis with an acoustic signal $A = [A_1, A_2, \ldots, A_N]$, sampled at fixed time intervals $1/f_s$ with a sampling frequency $f_s$. At this point we diverge from standard practices of machine learning in acoustics [39] (to use raw wave forms or frequency bank averages) and propose a new feature model. Described in detail in Sect. 4.4, it amounts to a nonlinear mapping $g(v)$ to a pair of time and amplitude sequences using an 'encoder+ denoiser', $A \xrightarrow{g(v)} (t, a)$. Here $t = [t_1, \ldots, t_{N_v}]$ and $a = [a_1, \ldots, a_{N_v}]$ are, respectively, time and amplitude sequences representing locations of 'relevant' acoustic events with a relationship $A(t_i) = a_i$; $v = (v_1, v_2)$ is a control parameter whose components are, respectively, the time and amplitude denoising thresholds. Finally, we use only half of this data through a new variable $x$,

$$x(v) = [x_1, x_2, \cdots, x_{N_v}], \qquad \text{defined by } x_i = -\log(t_{i+1} - t_i) - \log(f_s/2), \quad \text{parameterized by } v \tag{1}$$

as a single component feature, representing a local frequency estimate at $t(x_i) = t_i$. At this point, we have discarded the amplitude variable $a$ from further considerations.

For convenience, we will drop the explicit $v$ parameter notation, except where it clarifies the discussion. Now we introduce some required notation for data slicing and for relating the parameterized variable $x(v)$ and the physical time $t$. Any sub-sequence of the form $[x_i, \ldots, x_{i+L-1}]$ (of length $L$) is denoted as $x_{[i,i+L)}$. Any sub-sequence $x_{[i,i+L)}$ naturally induces a time interval, $I$, defined by the boundaries $I_{[i,i+L)} = [t(x_i), t(x_{i+L}))$. To simplify further the notation, we will use $x_r$ to specify a sub-sequence with a 'generic' range $r = [i, i + L)$, and likewise $I_r = I(x_r)$.

## 4.1 The statistical model

Our statistical model is a Bernoulli process ('coin toss') over events $E_I \in \Omega = \{\#(\text{impacts in } I) = 0, \#(\text{impacts in } I) > 0\} \sim \{0, 1\}$, with the probability

$$p(I_r(v)) = \mathbb{P}\{\#(\text{impacts in } I_r(v)) > 0 \mid \boldsymbol{\theta}\} = \sigma(h(\boldsymbol{x}_r(v); \boldsymbol{\theta}). \tag{2}$$

The log odds is modelled by a convolutional neural network (CNN) $h(\boldsymbol{x}_r; \boldsymbol{\theta})$ (defined in Sect. 4.5), and $\sigma(h) = 1/(1 + \exp(h))$ is the 'sigmoid' function, mapping $h$ to $[0, 1)$ interpreted as a probability. To summarize, the CNN operates on parameterized sequences $\boldsymbol{x}_r$ of various lengths (the minimal length is determined by a model's hyperparameters), and provides the probability that a physical impact occurs somewhere within a corresponding time interval $I_r$. We will adopt a standard decision rule parameterized by a cutoff value $c \in (0, 1)$. Namely, $E_{I_r} = 1$ if $\sigma(h(\boldsymbol{x}_r); \boldsymbol{\theta}) > c$ and $E_{I_r} = 0$ otherwise.

**4.1.1 Datasets from parameterized signal representations.** In the supervised learning context we provide a dataset consisting of pairs $\{(y_{r_i}, \boldsymbol{x}_{r_i})\}$ where the ranges $r_i$ are of fixed length, $|r_i| = L$. For this work we chose $L = 15$ as a compromise between the benefit of longer sequences for step identification and the benefit of shorter sequences for greater temporal precision. The feature $\boldsymbol{x}_{r_i}$ is a sequence of frequency estimates over a (typically short) time interval $I_{r_i}$, and $y_{r_i} \in \{0, 1\}$. The ranges are carefully determined to either sample from the gaps between annotated events (with 2 millisecond margins) for each $y_{r_i} = 0$, or to make sure that the annotated event is collocated near the center of a corresponding sequence, if $y_{r_i} = 1$. In this context, two data augmentations are used: 1) oversampling of $y = 1$ features by shifting the event location 4 positions around the center, and 2) strided sampling of the 'gap' features. These are dataset balancing measures, slightly improving heavily skewed datasets (see Table 2).

The features we are considering in a statistical model come from a parametric representation of the acoustic signal in which $v$ is an unspecified control parameter. A natural question is what value to use for the dataset. We will assume that our data is exposed to different parametric processes each possibly representing a different distribution of data points. To generate a dataset that explores the range of possible denoising parameters we create a union $\cup_{v, \boldsymbol{A}, r} \{y_r(v, \boldsymbol{A}), \boldsymbol{x}_r(v, \boldsymbol{A})\}$, where $v$ is sampled from some distribution for each available recording.

To ensure statistical correctness of the training and validation partitions, an 'arbitrary' time interval $I_{\text{valid}} = [10, 15)$ is reserved for the validation dataset, while the remaining available recording time is used for training. For simplicity, we chose an interval that each recording has and is not near the starting boundary. This approach guarantees that any 'impact' or 'gap' feature in *all parametric representations* appears in at most one of the partitions.

**4.1.2 Standard performance metrics.** The standard classification evaluation tools, where one class is underrepresented are the precision-recall curves (PR) and receiver operating characteristic (ROC) applied to the validation partition. The PR displays the precision, PPV = TP/(TP+FP) against the true positive rate (recall), TPR = TP/(TP+FN) for all available test cutoffs. Similarly, ROC displays TPR against the false positive rate 1−TNR, where the true negative rate is TNR = TN/(TN+FP). The area under curve (AUC) provides useful single measures for the overall model performance that is independent of the cutoff. In our case, ROC describes sensitivity to detecting the events, whereas PR describes the discriminating power. For example, a large ROC, but low PR values suggest that true positives are mostly identified, but a significant number of background is misidentified.

## 4.2 Time domain evaluation

Ultimately, we are interested in impact arrival time measurements, performed on contiguous recordings. In the inference mode, a fixed strided split of the recording into (overlapping) samples is used with a fixed stride $S$. It corresponds to taking $r_i = [Si, Si + L)$, $i = 0, \ldots$, where $S$ and $L$ are the stride and the sequence length, respectively. The nonlinearity of the acoustic feature model and sampling from multiple $v$ parameterizations complicate the picture in which fixed length features $r_i$ represent variable length time intervals $I_{r_i}$. Therefore, the standard dataset metrics do not directly address temporal precision-related questions.

**4.2.1 Cluster mean method: From interval to spike predictions.**   Cluster mean addresses the issue of translating interval predictions by Eq (2) into a sequence of impact time predictions and address two problems: mapping $I \mapsto t$ of an interval to a time prediction, and dealing with the possibility that multiple overlapping intervals may predict the same physical impact. It is relevant for short length features, when $\mathbb{P}\{\#(\text{impacts in } I) > 1\}$ is negligible for most $I$.

The first problem cannot be unambiguously resolved due to loss of information. We use a fixed offset approach: choose $K$ such that $0 \le K < L$ and posit $t(I_{[i,i+L)}) = t(x_{[i+K]})$. To address the second problem, we leverage the fact that a strided split of the form $x_{[Si,Si+L)}$ can also be viewed as a $i$-indexed sequence of intervals. Let $r = [i_1, i_2)$ be a range, in which all interval predictions are positive and are surrounded by at least one negative prediction on each side: $E_{I_i} = 1$ for all $i_1 \le i < i_2$ and $E_{I_{i_1-1}} = E_{I_{i_2}} = 0$. We identify such range with an impact, whose event time and the probability score are defined as

$$t_r = \frac{\sum_{j \in r} t(x_{Sj+K}) p(I_j)}{\sum_{j \in r} p(I_j)}, \quad p_r = \max_{j \in r}\left(p(I_j)\right). \tag{3}$$

The cluster mean is a convenient method of translating interval to event predictions. It allows to treat both the ground truth events and predictions on equal footing, as point processes (PP) which are ordered sequences of random numbers.

**4.2.2 Temporal pattern comparison based on fixed resolution scores.**   Precision of predictions is addressed by comparing a pair of given PPs $\mathcal{T}_{\text{ref}}$, $\mathcal{T}_{\text{test}}$ on an interval $I$, derived from respective empirical cumulative distribution functions. Typically one would be the annotations, the other—predicted impact times. We propose to measure their similarity as follows. Let $\epsilon > 0$ be fixed temporal precision, and let $B_{\mathcal{T}_i}(\epsilon)$ be a union of $\epsilon$-neighborhoods centered on each event in $\mathcal{T}_i$. Consider the fraction of events 'in the other process' that fall within $B_{\mathcal{T}_i(\epsilon)}$. There are two non equivalent alternatives:

$$p_1(\epsilon, v, c) = \mathbb{P}\left\{\mathcal{T}_{\text{test}} \in B_{\mathcal{T}_{\text{ref}}}(\epsilon)\right\} \tag{4a}$$

$$p_2(\epsilon, v, c) = \mathbb{P}\left\{\mathcal{T}_{\text{ref}} \in B_{\mathcal{T}_{\text{test}}}(\epsilon)\right\} \tag{4b}$$

In practice, $\mathcal{T}_{\text{test}}$ will be be determined by the parameterization $v$ and the cutoff $c$, which we have indicated as dependent variables. The temporal resolution $\epsilon$ is assumed small in comparison to the mean nearest neighbor distances of either PPs. $p_1$ and $p_2$ stand to each other in a similar relation as 'precision' stands to 'recall': either one, but not both, could be made arbitrarily close to 1 by making one of the PPs cover the interval $I$ with high enough density. Therefore, in analogy to the 'F score', we propose their harmonic mean as the objective function to maximize:

$$R(\epsilon, v, c) = \frac{2}{\frac{1}{p_1(\epsilon, v, c)} + \frac{1}{p_2(\epsilon, v, c)}}. \tag{5}$$

Point processes have a broad range of science and engineering applications; we have considered several popular distances and metrics including filtering-based metrics with Laplacian kernel [40, 41] and co-occurrence metric [42, 43]. They provide dimensionless ranking of predictions that are valuable, but do not directly address temporal resolution estimation, prompting us to propose the above methodology.

### 4.3 The main workflow: Training, validation and precision evaluation

The main steps of the workflow can be summarized as follows: pre-training on a dataset sampled from a 'broad' parameter distribution $\Phi_0$, followed by optimality study using scoring functions $R(\epsilon, v, c)$ and $p_1(\epsilon, v, c)$, and a final evaluation on a new 'narrow' dataset $\Phi_1$. These steps will lead to understanding of the role of $v$, temporal precision and suggest a real time inference strategy.

As a first step, each recording with available annotations is encoded as described in Sect. 4.4, using a sample of $v$ taken from $\Phi_0$, defined as

$$\text{'Broad'} \quad \Phi_0(v) = \{v_1 \sim \mathcal{U}(F_{\tilde{\delta t}}^{-1}(0.5), F_{\tilde{\delta t}}^{-1}(0.99)), \\ \ln v_2 \sim \mathcal{U}(\ln F_{|\delta \tilde{a}|}^{-1}(0.5), \ln F_{|\delta \tilde{a}|}^{-1}(0.99))\} \, . \tag{6}$$

Here $\mathcal{U}(a, b)$ denotes uniform random variable between $a$ and $b$, $F^{-1}(0.5)$ and $F^{-1}(0.99)$ are, respectively, the median and the 99-th percentile with respect to the relevant variable. The latter are the parameter-free encoding deltas, discussed in Sect. 4.4 (see also Fig 3, left pane). Since the dataset size scales with the number of samples, 10 samples are used to generate $\Phi_0$. It is probable that none of them are nearly optimal, and some are quite poor.

The negative log likelihood amounts to the standard logistic regression loss function

$$\text{Loss}(\theta) = \sum_v \sum_r \text{CrossEntropy}(y_r(v), p(I_r(v))) \, . \tag{7}$$

with averaging over $v$ included. The theoretical goal is

$$\hat{h}(\cdot) = h(\cdot; \hat{\theta}) \, , \quad \text{where} \quad \hat{\theta} = \arg \min_\theta \text{Loss}(\theta) \, . \tag{8}$$

In practice, we search for a minimum of Eq (7) using the stochastic gradient descent (SGD)

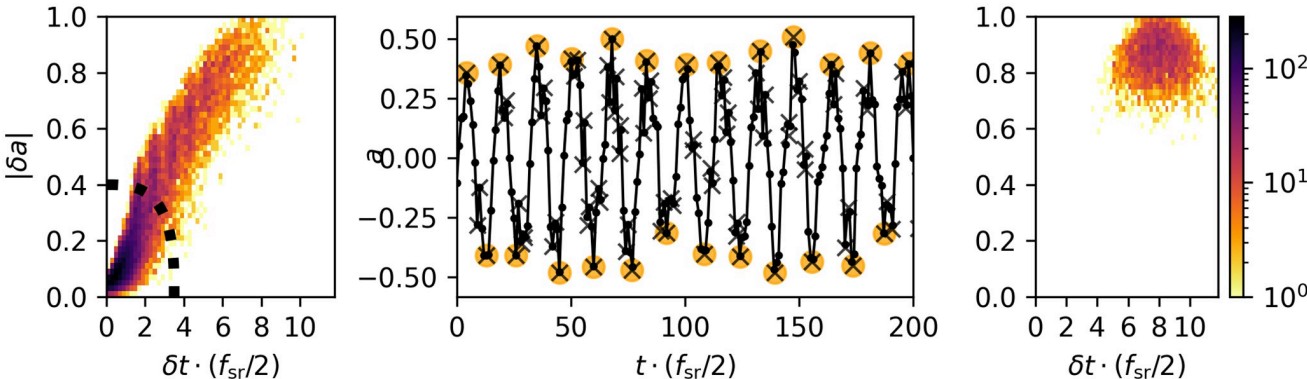

**Fig 3. Acoustic feature model: Differences between encoding and denoising.** Left: Distribution of $\delta \tilde{t}$, $|\delta \tilde{a}|$ shows a large concentration of noise near the origin. A boundary $B_v$ (thick curve) will be used to define noise. Right: Distribution of $\delta t$, $|\delta a|$ after denoising shows a cleaned up signal and a partially recovered 1kHz frequency. Middle: a sample of the signal showing the acoustic events. Connected dots: original sampling, crosses: encoded $(\tilde{t}, \tilde{a}) = g_0(A)$, large circles: denoised $(t, a) = g(v)(A)$.

numerically [14] using several trials and a fixed number of iterations. Model performance is evaluated as described in Sect. 4.1.2 on the validation partition and the best model $\hat{h}$ is selected using an additional score, e.g. PR AUC.

The hope is to obtain a fault-tolerant model $\hat{h}$ that generalizes to unseen data, but the premise may look ill conceived: The model is trained on an averaged $\Phi_0$ (confirmed by modest training performance) and there is no guarantee that any specific $v$ should perform better than the average. To investigate this important effect the tools of Sect. 4.2 will be valuable. The high road is to study optimality of the scores $R(\epsilon, v, c)$ or $p_i(\epsilon, v, c)$ with a fixed $\epsilon$ as a function of $v$, e.g.:

$$\hat{R}(\epsilon, v) = R(\epsilon, v, \hat{c}), \tag{9a}$$

$$\begin{aligned}\hat{p}_1(v) &= p_1(\epsilon; v, \hat{c}),\\ &\text{where } \hat{c}(v) = \arg\max_c R(\epsilon, v, c) \text{ and } \epsilon = 0.01,\end{aligned} \tag{9b}$$

since they are informative about the percentages of hits and misses with $\epsilon$ precision. This leads to a better understanding of a 'score topography' and allows to locate the best $v$ for a given score function. However, since $v$ and $c$ have a different standing: $v$ is a control parameter, but $\hat{c}$ is a result of optimization, a more robust estimate is of interest. This argument motivates to consider $v^*$, defined by

$$v^* = \arg\max_v \mathbb{E}_{c \sim \mathcal{U}[0.5, 0.8]} R(\epsilon, v, c), \tag{10}$$

which is optimal 'on average' over a typical range of cutoff, and would not require the prior knowledge of its optimal value. Other statistics such as median or quantiles could be used, and yield similar results.

With the knowledge of a tentative best value $v^*$ we propose to generate a 'narrow' dataset $\Phi_1(v)$ which is equivalent of $v^*$+ noise.

$$\begin{aligned}\text{'Narrow'} \quad \Phi_1(v) = \quad &\{3 \times 3 \text{ grid, centered on } v^* :\\ &v_1 \in v_1^* + [-1, 0, 1] \cdot (2/f_s),\\ &v_2 \in v_2^* + [-1, 0, 1] \text{ dB}\}\end{aligned} \tag{11}$$

Although not strictly necessary, the last step shows how to quantify the optimality by comparing $\Phi_0$ and $\Phi_1$ using dataset-based performance evaluation tools.

## 4.4 Acoustic feature model

The basic idea of the proposed audio feature model is to track the relevant local peaks and troughs of the signal $A(t)$, recording both the extremal amplitude values, $a_i$, and their temporal locations $t_i$. In the first step, we track *all* local extrema with a parameter-free *encoder* $g_0$: $(\tilde{t}, \tilde{a}) = g_0(A)$, with implied relations $\tilde{a}_i \approx A(\tilde{t}_i)$ and $A'(\tilde{t}_i) \approx 0$. This encoder can be implemented using bit-shift and comparison operations and used as a low latency audio processing component (see Algorithm 1). A quadratic polynomial interpolation based on three adjacent nodes was adapted to refine the locations of the extrema.

**Algorithm 1 Encoder pseudo code** '$t$' and '$b$' are memory cells, recording local 'top' and 'bottom' conditions, respectively. Incremental time is recovered from the differential $\text{encT}_j$ and the sampling frequency $f_s$. An optional improvement step uses a quadratic interpolation between the three values straddling a local extremum.

```
1: function ENCODE(input, encoded)
2:    INITIALIZE(t, b, steps, i, j)
```

```
 3:    current ← input_i
 4:    while MoreData(input, encoded) do
 5:      do
 6:        i ← i + 1
 7:        next ← input_i
 8:        steps ← steps + 1
 9:      while next = current              ▷ Addressing the ties
10:        t ← (ShiftLeft(t, 1) | (current < next)       ▷ 'top' marker
11:        b ← (ShiftLeft(b, 1) | (current > next)       ▷ 'bottom' marker
12:        if (t & 0x03 = 0x01) | (b & 0x03 = 0x01) then    ▷ Extremum
    found!
13:          encA_j ← input_{i-1}
14:          encT_j ← steps - 1         ▷ t_j = t_{j-1} + \frac{1}{f_s} encT_j
15:          Interpolate(encA_j, encT_j | input_{i-2}, next)   ▷ Optional step
16:          encoded_j = (encA_j, encT_j)
17:          steps ← 1
18:          j ← j + 1
19:        end if
20:        current ← next
21:      end while
22: end function
```

A more realistic model requires handling the electronic noise and undesirable environmental sounds that may be mixed into the target signal $A(t)$. We propose a heuristic denoiser $g_1(v)$ aiming to mitigate high to moderate signal to noise ratio situations. It is based on the notions of outliers and 'noisy sequences' with respect to the delta variables $\delta \tilde{t}_j = \tilde{t}_j - \tilde{t}_{j-1}$, and $\delta \tilde{a}_j = \tilde{a}_j - \tilde{a}_{j-1}$. The main assumption is that the distribution of $\delta \tilde{t}$ and $|\delta \tilde{a}|$ is concentrated near small values, and is mostly noise, whereas the signal of interest is represented by 'rare' and large excursions. To approximately subtract the noise from the total distribution we use a heuristic boundary function $B_v$, parameterized by a denoising control parameter $v = (v_1, v_2)$. For example, an 'oval' boundary and $v$ are defined so that

$$B_v(x, y) = \left(\frac{x}{v_1}\right)^3 + \left(\frac{y}{v_2}\right)^3, \tag{12}$$

and $B_v(\delta \tilde{t}_i, |\delta \tilde{a}_i|) < 1$ implies that $t_i$, $a_i$ are part of a noisy sequence. Data that satisfies such condition typically form sequences, which we simply eliminate. The full acoustic feature model is a composition $g(v) = g_1(v) \circ g_0$.

**Example: pure signal** Let $A(t) = \sin(2\pi f t)$ be sampled at a rate $f_s$. Then $A = \{\sin(2\pi f n / f_s)\}_{n=1}^{f_s T}$. Denoising has no effect: $t = \tilde{t} = \{\frac{n}{2f} + \mathcal{O}((f/f_s)^{\xi+1})\}_{n=1}^{fT}$, and $a = \tilde{a} = \{(-1)^n + \mathcal{O}((f/f_s)^\xi)\}_{n=1}^{fT}$; $\xi = 2$ for a quadratic interpolation, $\xi = 1$ for no interpolation. Note that the result may have considerably fewer samples than the original. Lastly, the vector $x$, defined by Eq (1), is constant, $x_n = \log f - \log(f_s/2)$.

**Example: noisy signal** Consider a sinusoid with $f = 1$ kHz, $f_s = 16$ kHz mixed with 5dB white noise. Fig 3 shows the encoded $\tilde{t}$, $\tilde{a}$ distribution, the boundary to separate noise from the signal (left pane), and the distribution after denoising (right pane). The middle pane illustrates the actual 'events' that we're focusing on, i.e. the local extremes of the amplitude profile. This example shows that the denoiser $g(v)$ can approximately recover the frequency of a contaminated signal, making it an effective first defense against high-frequency noise encountered in practice.

There is longstanding acoustic feature research for music, human voice and animal sound classification. To our knowledge, the present model has not been previously proposed. To an

extent, it is analogous to the 'zero crossings' method, if the crossings were applied to the derivative $A'(t)$ instead of $A$. The analogy does not extend further both in terms of implementation as well as use cases: zero crossings are biased toward low frequencies, whereas our AFM is sensitive to high frequency detail and, in addition, we apply denoising. It is interesting to note that zero crossings has been previously proposed for percussive sound classification in contemporary music [44].

## 4.5 Compact CNN architectures for improved impact detection

Our goal in using machine learning is to improve the noisy local frequency estimates provided by the feature $x$. The framework of choice is one-dimensional artificial convolutional neural network (1D-CNN).

The rain drop counting is posed here as an object (event) detection supervised machine learning task, where each physical impact is counted as an event. What sets this problem apart from other acoustic event or scene identification tasks [45, 46] are very short duration, almost instantaneous events. A further complication is that the relaxation dynamics conceals the trailing edge of a physical impact, leaving only the leading edge and very little temporal depth to go by for developing a pattern recognition strategy. All considered CNN architectures (illustrated in Fig 4) have a 'denoiser-detector-resolver' structure. The 'resolver' consists of a global max pooling layer (GM), one or more fully connected layers (FC), and a softmax unit for binary classification. The GM implements a 'best of' logic and erases the event location within a sequence, but allows inputs of various lengths to be used. This 'resolver' part of the model is fixed (except for the hyperparameter values).

**4.5.1 Basic detector model.** The basic idea of a 'step detector' is to identify the leading edge of an impact with a step within a sequence $x_{[i, i+L)}$, and then to use 'step filters' to detect them as sufficiently large peaks. The step filter is implemented as a convolutional layer with $C$ filters, resulting in $C$ output channels. To improve the initial guess, we use a non-standard weight initializer imitating a random step function (see the 'random' pane in Fig 4, right column).

The main reason to investigate more complex models is that the basic model is susceptible to high frequency noise, whose common cause is wind. The detector is still considered a useful component, implemented in the subsequent models as a *separable* convolutional layer, and

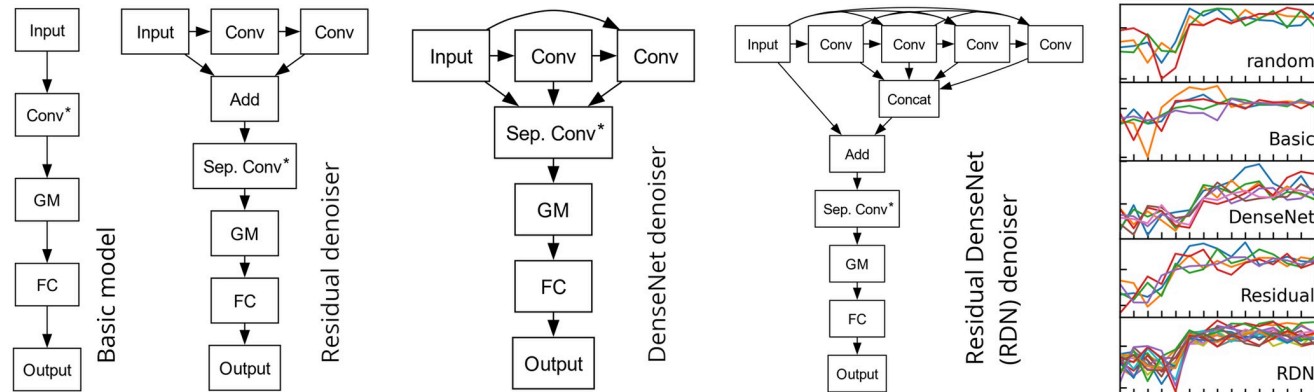

**Fig 4. Comparison of 1D-CNN architectures.** Residual, DenseNet, and RDN denoising units explore convolutional topologies for high-frequency noise suppression. Conv: Convolution layer followed by batch normalization and ReLU activations. GM: global max, FC: fully connected layers; Step detector: Conv* and Sep. Conv* (separable convolution with unit depth multiplier). Add: adds 'Input' to each remaining channel. Multiple inputs to all Conv layers are concatenated. Plots: samples of a random Conv* initializer and trained model filters.

**Table 1. Summary of 1D-CNN model parameters, complementary to Fig 4.** Denoisers: channels of respective Conv layers; Input and output channels of a step detector; FC: units of a fully connected layer.

| Model | Denoiser layer parameters | Step detector | | FC layer outputs | Trainable parameters |
|---|---|---|---|---|---|
| | | inputs | outputs | | |
| Basic | N/A | 1 | 5 | 12 | 165 |
| Residual | $C_1 = 6, C_2 = 5$ | 5 | 5 | 12 | 464 |
| DenseNet | $C_1 = C_2 = 3$ | 7 | 7 | 12 | 738 |
| RDN | $C_1 = C_2 = C_3 = C_4 = 4$ | 17 | 17 | 16 | 2305 |

marked with an asterisk in Fig 4, to distinguish the non-standard weight initializer. One perceived disadvantage of the basic model is the use of a single input channel. The following models could be interpreted as a study to generate more input to the detector in a way that produces useful context for denoising.

**4.5.2 Models with additional denoising layers.** The deep residual network architecture (ResNet) is well known in computer vision [47] and has also been applied to image denoising [48]. In a **model with residual denoiser**, we use a single residual block $\mathcal{R}$ consisting of two convolutional layers with $C_1$, $C_2$ channels, respectively, adapted to a 1D-CNN setting. The input is channel-wise added to each output channel of $\mathcal{R}$, therefore the input to the detector layer is of the form $\boldsymbol{x}_{[i,i+L]} + \lambda \mathcal{R}_j(\boldsymbol{x}_{[i,i+L]}), j = 1, \ldots, C_2$. The parameter $\lambda$ is controlled by the weight regularization of the convolutional layers of $\mathcal{R}$.

In a similar fashion, a **model with DenseNet denoiser** uses a single DenseNet block [49], combining $i = 1, 2$ convolutional layers with $C_i$ channels respectively and a DenseNet connectivity pattern. Since the the DenseNet block concatenates all combinations of its layers, the detector input has $1 + C_1 + C_2$ channels.

The Residual DenseNet architecture, which combines ResNet and DenseNet topologies, has been proposed for image denoising [50]. Similarly in a **model with Residual DenseNet (RDN) denoiser** we employ a single RDN block, consisting of four convolutional layers each with $C_i$ channels, suitably adapted to 1D-CNN setting.

All model topologies are summarized in Fig 4, and their hyperparameters are shown in Table 1. The kernel size is fixed at $K = 15$ for all models, which is equal to the dataset sequence length ($K = L$). All considered models are applicable when $L \geq K$. Where applicable, L2 regularization is used. The hyperparameters are chosen so as to make all models have similar extents.

# 5 Results and discussion

## 5.1 Contribution of data

We committed a few recordings with annotated impacts to a new dataset. The physical impacts are identified with 1 millisecond bounding boxes, annotated in practice to three decimal digits. Higher resolution is possible but not required. Recordings have been collected in urban environment (northern Italy) with a smartphone with controlled gain and 48kHz sampling rate, downsampled to 44100 kHz for further analysis. The devices were not shielded from wind or environmental noise. For example, recording with ID 1 is marked as containing thunders, 2—street noises, 3—strong wind.

Annotating such recordings is a highly laborious task since ideally each impact should be identified, sometimes in the presence of strong background noise (see Fig 1). The methodology we found to work best is to play them at 5%–10% of the original speed implementing,

**Table 2. Summary by recording: Annotations, training features, and optimal parameters.** 1) Recordings: Len—Total recording time in seconds, $N$—Number of annotated events, $\tau$—mean inter arrival time (ms), $\lambda$—arrival rate per unit area ($s^{-1}m^{-2}$); 2) $\Phi_0$-sampled training data: $n_1$—#$\{y = 1\}$, $n_0$—#$\{y = 0\}$; 3) Optimal estimates, $v^*$—Eq (10), $\hat{c}^* = \hat{c}(v^*)$, $\hat{R}^* = \hat{R}(v^*)$, and event counts $\hat{N}^*$.

| ID | Len | $N$ | $\tau$ | $\lambda$ | $n_1$ | $n_0$ | $v^*$ | $\hat{c}^*$ | $\hat{R}^*$ | $\hat{N}^*$ |
|---|---|---|---|---|---|---|---|---|---|---|
| 1 | 20 | 208 | 95 | 1426 | 1539 | 15602 | (5,19) | .58 | .87 | 371 |
| 3 | 34 | 118 | 294 | 461 | 932 | 20865 | (4,13) | .72 | .78 | 201 |
| 5 | 50 | 110 | 457 | 296 | 960 | 44425 | (4,26) | .81 | .69 | 373 |
| 4 | 107 | 734 | 146 | 928 | 7015 | 88572 | (3,27) | .69 | .86 | 1447 |
| 2 | 191 | 705 | 271 | 499 | 6840 | 162963 | (3,24) | .73 | .88 | 1540 |
| Totals | 402 | 1875 | | | 17286 | 332427 | | | | |

essentially, a downconverter from 20 kHz frequency range to a well audible range where an impact's 'thump' is very clear; for that we used the 'Audacity' software [35]. Tentative impacts were cross-examined visually, using an amplitude profile and 'feature' plots such as those, shown in Fig 1. Generally we followed a conservative policy: 'do not annotate if in doubt'. The result so far is 1875 unique events from five recordings, summarized in Table 2.

## 5.2 Model training and validation on datasets

Models are trained exclusively on the $\Phi_0(v)$-sampled dataset, see Eq (6). The initial part of the training progression is illustrated in the right column of Fig 5. All models are optimized to a class-weighted binary cross entropy using the stochastic gradient descent (SGD) with variable learning rates and mini batches of 512 or 256 samples [51]. The class weighting is necessary because of highly imbalanced training dataset (see Table 2). Training was restarted multiple times with random initial conditions and trained for 150 or 300 epochs. The denoising layer weights (upstream of the respective step detectors) were $L_2$ regularized with a parameter 0.01 and batch-normalized (with the exception of the basic model that does not use denoising). No other regularization measures were applied. The prediction-recall and receiver operating characteristic areas under the curves (PR AUC and ROC AUC) at the end of each training cycle, and their rms values are summarized in Table 3. The curves (best PR case) are shown in Fig 5, middle panes. Validation datasets with $\Phi_0$ (above) and $\Phi_1$ (below) are confronted for the same models (which were trained on the $\Phi_0$ dataset).

The training progressions in Fig 5 suggest that the general tendency is to underfit. A 'dip', seen in some early stages of progression is explained by the initial weight distribution of the detector layer, which presumably identifies both true and false frequency jumps, so that most of the remaining iterations are burdened with reducing the number of false positives.

## 5.3 Search for optimal control parameters

Next, we evaluate Eqs (9a), (9b) and (10) on individual recordings with $\epsilon = 0.01$, using a $\Phi_0$-trained residual model $\hat{h}$ from Sect. 4.5.2 as a case study. Available recordings are parameterized on a $10 \times 10$ $v$-grid, with $v_1 \cdot f_s/2$ and log $v_2$ spanning, respectively [1, 10] and [11, 38] dB. Cluster mean predictions, Eq (3), are generated for a range of 70 cutoffs $c \in [0.2, \ldots, 0.9)$, yielding 7000 predictions per recording. Their comparison with corresponding annotations yield $p_1(\epsilon, v, c)$, $p_2(\epsilon, v, c)$ and $R(\epsilon, v, c)$. First, $\hat{R}(\epsilon, v, \hat{c}(v))$ and $\hat{p}_1(\epsilon, v, \hat{c}(v))$ are computed by a straightforward application of Eqs (9a) and (9b) and visualized in Fig 6, panes 2–6: as density maps of $\hat{R}(\epsilon, v, \hat{c}(v))$, and as contour lines of $\hat{p}_1(\epsilon, v, \hat{c}(v))$, overlaid. Then, $c \in [0.5, 0.8]$ is used to compute the mean in Eq (10) and to find $v^*$, which is listed in Table 2. Fig 6, pane 1 shows

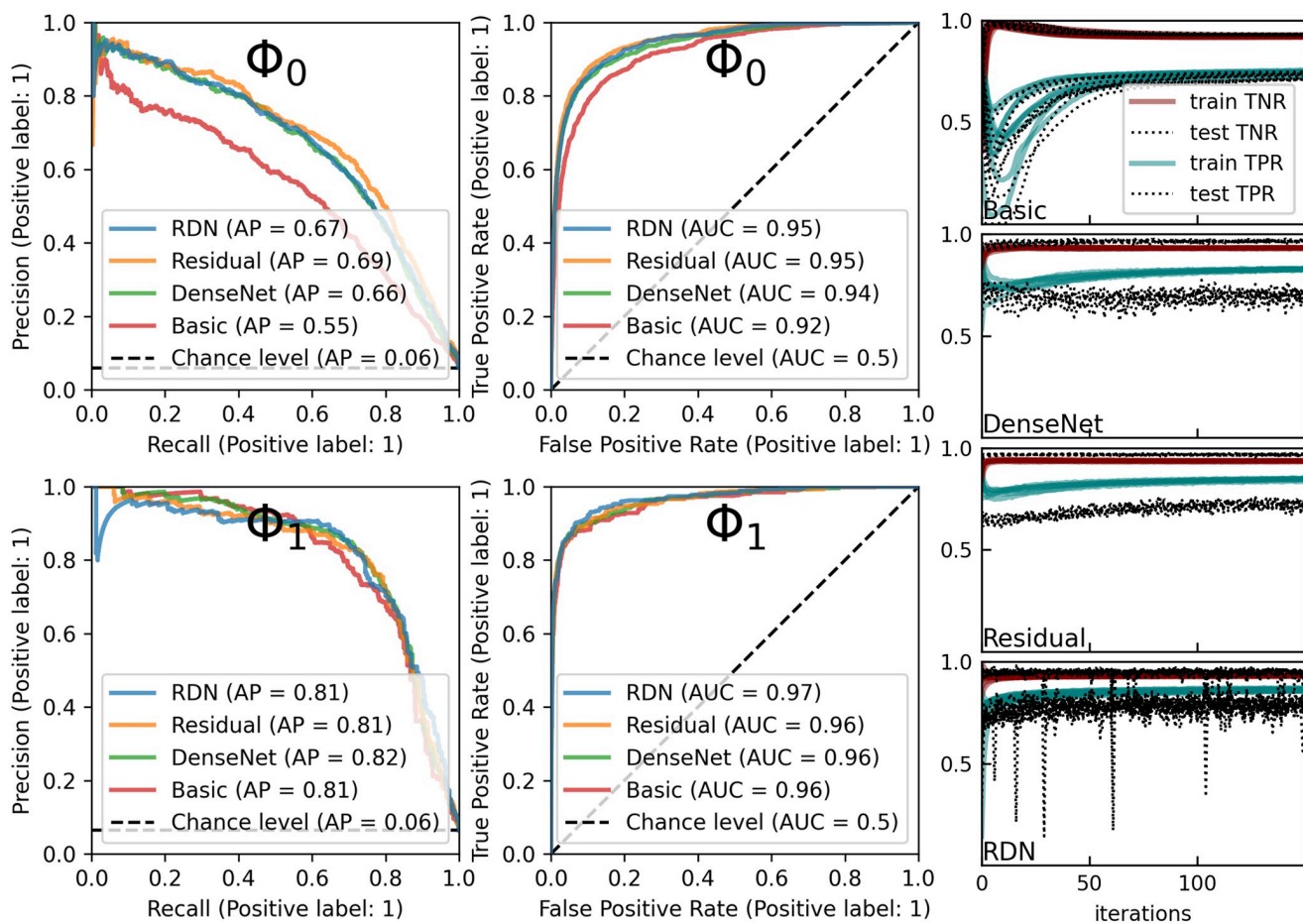

**Fig 5. Training progress on $\Phi_0$ dataset and validation performance on $\Phi_0$ vs $\Phi_1$.** Right column: initial training progress showing multiple sample runs. Left four panels: validation performance comparison on $\Phi_0$ and $\Phi_1$ datasets for the models, trained on $\Phi_0$ dataset.

the $p_1(\epsilon, v^*, c)$ against $p_2(\epsilon, v^*, c)$ for all available cutoff values $c$. Finally, Fig 7 illustrates a sample for the optimal $v^*$, namely $p(\epsilon, v^*, \hat{c}(v^*))$ for a range of $\epsilon$ thresholds up to 30 ms (left pane), the predictions in the validation range (middle column), and the cumulative probability distribution of the first arrival time, $F_\mathcal{T}(t) = \mathbb{P}(t_{i+1} - t_i < t)$, which is an example of statistics afforded by the present methodology (right column). A considerable fraction of predictions was found to be paired at less than 10 ms. The red line shows the full prediction distribution, whereas the green line shows the distribution a subset filtered by $\{t_{i+1} - t_i \geq 0.01\}$.

**Table 3. Summary of model performance.** Complementary to Fig 5 summary of PR and ROC AUC for $\Phi_0$ and $\Phi_1$ validation datasets, showing respective mean and rms (in parentheses).

| model | $\Phi_0$ | | $\Phi_1$ | |
|---|---|---|---|---|
| | ROC × 100 | PR × 100 | ROC × 100 | PR × 100 |
| Basic | 85.9 (0.6) | 38.8 (1.7) | 90.7 (0.7) | 67.4 (1.1) |
| DenseNet | 86.3 (0.5) | 49.3 (2.6) | 89.6 (0.8) | 71.7 (1.4) |
| Residual | 86.9 (0.3) | 52.8 (0.7) | 89.3 (0.7) | 71.1 (0.9) |
| RDN | 88.2 (1.6) | 47.9 (3.5) | 90.1 (1.4) | 67.4 (4.8) |

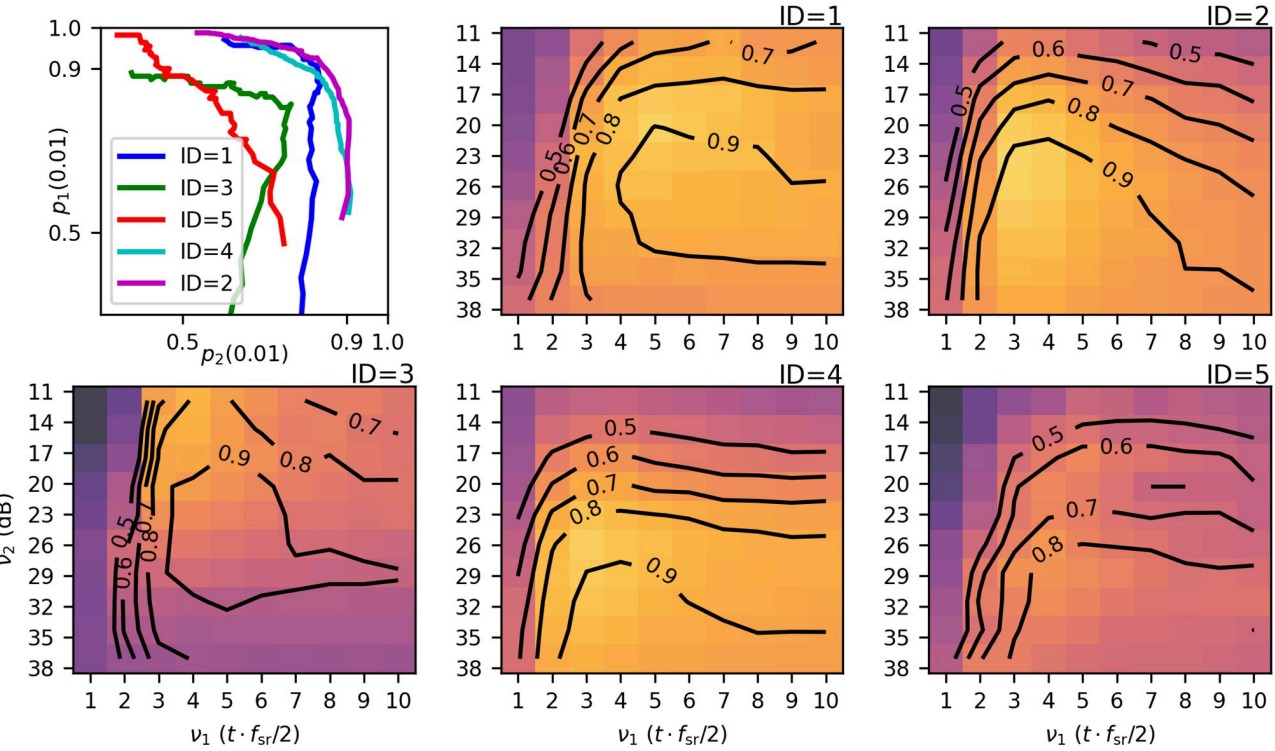

**Fig 6. Landscape of scores over a grid of denoising parameters, panes 2–6, for each recording.** Color: $R(\epsilon, v, \hat{c}(v))$ (lighter is better). Contour lines: $p_1(\epsilon, v, \hat{c}(v))$ (higher is better). Pane 1: $p_1(\epsilon, v^*, c)$ vs $p_2(\epsilon, v^*, c)$ for all available cutoff values $c$.

### 5.4 Discussion

**5.4.1 Data considerations.** It is of interest to include additional data to cover extreme conditions (large and small drop size limits, high arrival intensity) as well as explore material properties and their temporal degradation. Note that a better ROC vs PR performance in Fig 5 suggests that improving high frequency denoising is more relevant than collecting more impacts, for which the acoustic feature model is quite effective.

The first outstanding question is therefore how to efficiently generate more reliable impact annotations. Natural rain recordings are easy to obtain but the labelling is difficult and possibly prone to bias because the audible signature (percussive bursts) and detection principle (steps in local frequency) are of the same origin. Manual precision dispensers could be used to create individual drops of controllable dimensions. Simultaneous recording with high-speed camera could provide valuable independent cross-validation. Both these methods are feasible only for short time spans with a caveat that their own data would have to be properly interpreted.

Notice that our annotation policy is biased to marking only highly plausible impacts. Almost certainly, we failed to mark a few true impacts with 'weaker' signatures (among possible causes being high noise, drop-on-drop impacts, very small droplets). Our data should be viewed from the perspective of label noise with an asymmetric distribution (more unlabelled impacts than incorrectly labelled noise), or as data that contains positive and unlabelled data (PU learning). Since both are mature branches of machine learning, the most promising direction to efficiently generating new annotations or improving the existing ones, and reducing the label noise, may be to adapt an appropriate methodology from reviews [52, 53].

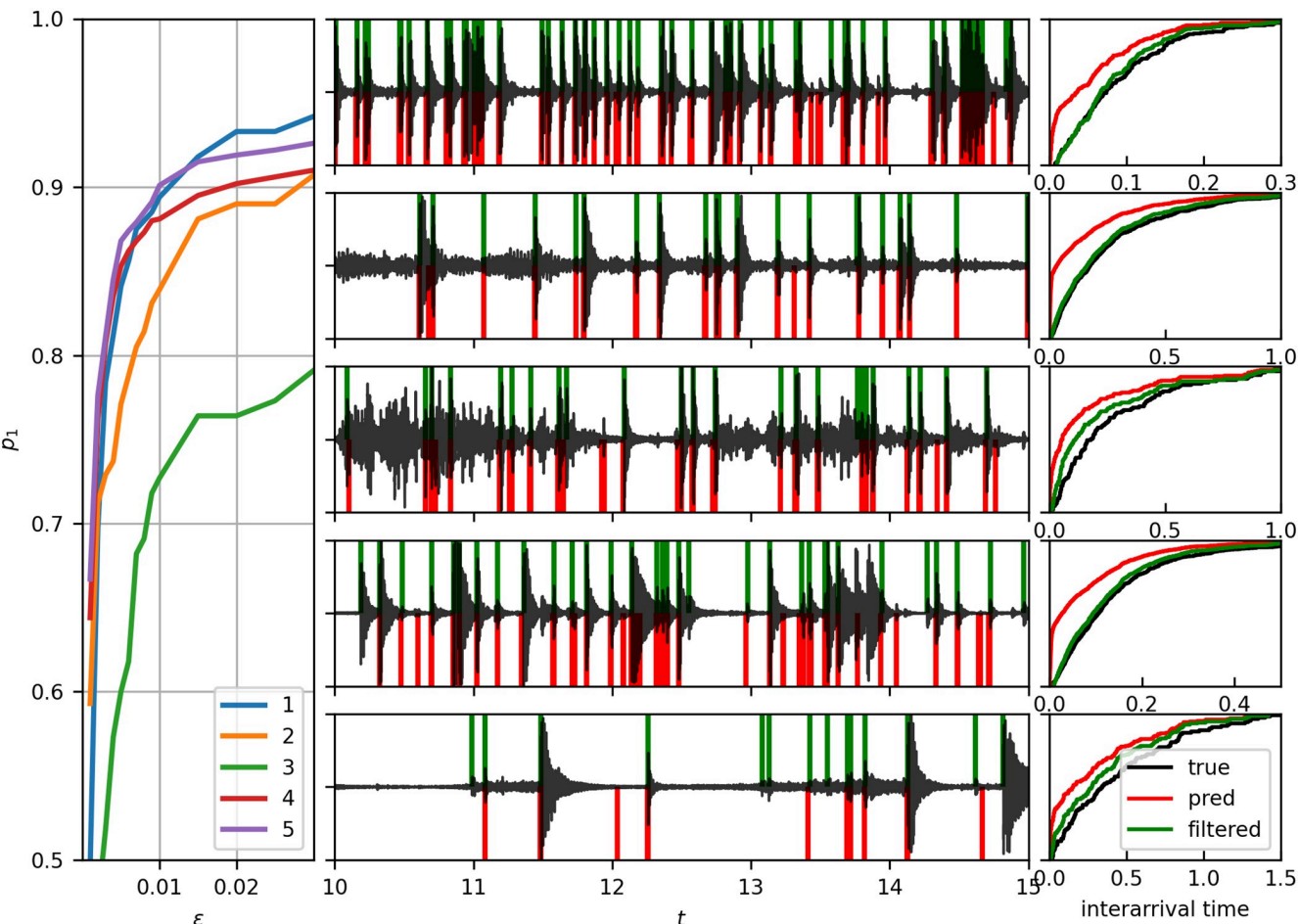

**Fig 7. Selected sample predictions.** Left: fraction of predicted events $p_1(\epsilon, v^*, \hat{c}(v^*))$ $\epsilon$-close to the ground truth, Eq 4a, for various $\epsilon$ and each recording. Middle: audio signal in the validation range and ground truth (green verticals, above zero) vs predictions (red verticals, below zero). Right: the corresponding CDFs. Red line: raw CDF, green line: predictions with spurious duplicates removed.

Lastly, Figs 6 and 7 (left column) identify the recording ID = 5 as an outlier, although it should have been an 'easy' case. It is suspected that a defective sensing surface lost tension, possibly following a deformation of the outer casing.

**5.4.2 A blueprint for a real time application.** The second outstanding question is how to obtain useful models and denoising parameters for a real time application. We believe that a blueprint for a solution lies within the previous sections: Ample evidence is provided by the $\Phi_0$–$\Phi_1$ comparison (Fig 5 and Table 3) and landscapes of Fig 6 that $\Phi_0$-trained models are highly selective. This implies that a suitable pre-trained model will exceed average performance in a broad range of conditions if the denoising parameter is attuned to these conditions. A possible way forward for performing such selection is to turn a larger number of covariates $\tilde{\delta t}, |\tilde{a}|, v^*$ from short time intervals as well as recordings without rain into a predictive model. Such a model would perform dynamic calibration by estimating $v^*$ for the current time interval from the past data.

**5.4.3 Model training lesson learned.** We found that sampling from a broad distribution such as $\Phi_0$ is also one of the main ingredients to successfully train models that generalize to unseen data, the second being a sufficiently large batch size. A large batch size (e.g. 256) is

required to reasonably guarantee that a few positive samples are included in each each mini-batch. Our experiments show that the workflow is not iterative: retraining on $\Phi_1$ fared worse on the generalization test. We speculate that a broad distribution ($\Phi_0$) sampling mitigates the sharp minimizer problem [54] associated to large batch training.

**5.4.4 Model differences.**   The primary goal of investigating several CNN architectures with denoising layers is to mitigate high-frequency noise, usually caused by wind, for which the basic model seems too simple. The $\Phi_0$ vs $\Phi_1$ PR and ROC comparison is interesting, suggesting that all models perform well on optimal parameters, but the denoising models perform better on average. It remains to find substantial arguments in favor of either model in Sects. 4.5.2.

**5.4.5 Temporal and event count precision.**   The left pane of Fig 7 shows $p_1(\epsilon, v^*, \hat{c}(v^*))$ as a function of $\epsilon$, interpreted as the fraction of predictions that are within $\epsilon$ to the ground truth. It follows that 80–90% of predictions lie within 10 ms in all cases except for the outlier ID = 5. This observation is the basis of our initial claim of accuracy.

The event count density is systematically greater with respect to ground truth, as seen in Table 2, but also indirectly in Fig 6, pane 1, showing prevalence of $\hat{p}_2 < \hat{p}_1$. As suggested by Fig 1 case D, some extraneous predictions are in fact plausible true events that have not been marked, i.e. they represent corrected label noise. The numerically largest fraction of spurious predictions come from an artifact most likely due to the cluster mean prediction algorithm. It consists in predicting pairs of events a few milliseconds apart for a single reference event. This is demonstrated using cumulative distribution functions (CDFs) in Fig 7, right column. Filtering out events at $<$10ms yields much better CDFs. It is believed that improvement of the cluster mean method and, possibly, CNN hyperparameters will solve this issue in the future. The cause of most concern are false positives due to wind, since strong gusts can excite deceptively similar acoustic signatures. Wind as a source of inaccurate readings due to shaking has been identified with traditional impact disdrometers as well. Possible remedies include wind shielding, experimenting with thicker materials, or adopting more effective denoising methodologies.

**5.4.6 Generalizations and extensions.**   An interesting generalization to pursue is an acoustic equivalent of algorithm unrolling. The goal would be to convert the current methodology, composed of several isolated components, to an equivalent, end-to-end deep learning workflow. The paradigm of algorithm unrolling [55] has been successfully used in image denoising and, importantly, it presents a path to embed known fundamental facts in an explainable deep learning modelling [56]. Explainable models are easier to generalize to other settings. For example, it is of interest to extend these results to the physics of crumpling thin sheets, which has recently attracted considerable attention [57] with a wealth of experimental and theoretical [58, 59] results available.

# 6 Conclusions

High precision instrumentation usually implies materials of superior quality and fine manufacturing processes involved. It is of interest when high precision can be achieved with off-the-shelf materials and coarse 'manufacturing'. This type of innovation is the main theme of the present work, focusing on precision rain drop counting with an ultra low cost rig, a new detection principle and innovative signal analysis. Our main proposal is based on frequency detection of a high frequency 'percussive' acoustic transient, using a drum-like sensor, which is an unfamiliar detection principle for rainfall measurement instrumentation. The origin of this transient is unclear and invites further experimentation with soft hydrophobic materials. In a 'proof is in the pudding' approach, we demonstrate the feasibility with a concrete

numerical implementation and a detailed analysis of the results. From signal analysis perspective we are seeking to identify percussive impacts in a noisy signal, leveraging time scale separation. Our implementation has two original components: a non-local audio processing unit that we call 'acoustic feature model' and a convolutional neural network architecture, optimized to deal with noise and to improve the predictions. This methodology is designed with an explicit goal to run 'near the source', on small, resource constrained microcontrollers or smartphones. It will be of interest to a growing community of embedded engineers and TinyML enthusiasts, because it is compatible with embeddedML ready [10] microcontrollers. As a meteo-instrument, the hardware is limited by the durability standard. While there are also implementation details to be optimized (such as improving the method of translating interval predictions to event counts) and field testing in real time conditions to be performed, we nevertheless believe this methodology to be of interest in certain environmental sensing applications, especially where ultra low budget and short term campaigns are involved. We also believe the methodology to have broader interest in applications where detection of percussive events is of interest, from music classification to the crumpling of thin sheets.

## Acknowledgments

Marco Zennaro (ICTP) is acknowledged for useful insights. The Information and Communication Technology Section is acknowledged for providing access to the EuroHPC supercomputer LEONARDO [60], hosted by CINECA (Italy).

## Author Contributions

**Conceptualization:** Rytis Paškauskas.

**Data curation:** Rytis Paškauskas.

**Formal analysis:** Rytis Paškauskas.

**Funding acquisition:** Rytis Paškauskas.

**Investigation:** Rytis Paškauskas.

**Methodology:** Rytis Paškauskas.

**Project administration:** Rytis Paškauskas.

**Resources:** Rytis Paškauskas.

**Software:** Rytis Paškauskas.

**Supervision:** Rytis Paškauskas.

**Validation:** Rytis Paškauskas.

**Visualization:** Rytis Paškauskas.

**Writing – original draft:** Rytis Paškauskas.

**Writing – review & editing:** Rytis Paškauskas.

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
