## [Decision Letter · Decision Letter 0]

6 Aug 2024

PONE-D-24-29859High-resolution raindrop counting via instantaneous frequency sensing on hydrophobic elastic membranesPLOS ONE

Dear Dr. Paškauskas,

Thank you for submitting your manuscript to PLOS ONE. After careful consideration, we feel that it has merit but does not fully meet PLOS ONE’s publication criteria as it currently stands. Therefore, we invite you to submit a revised version of the manuscript that addresses the points raised during the review process.

**ACADEMIC EDITOR: **The manuscript introduces a positive results and novel method for high-resolution raindrop counting using instantaneous frequency sensing on hydrophobic elastic membranes. it can be accepted in PLOS ONE after minor revision.==============================

We look forward to receiving your revised manuscript.

Kind regards,

Zaky A. Zaky, Lecturer

Academic Editor

PLOS ONE

Journal Requirements:

Reviewers' comments:

Reviewer's Responses to Questions

**Comments to the Author**

1. Is the manuscript technically sound, and do the data support the conclusions?

Reviewer #1: Yes

Reviewer #2: Yes

2. Has the statistical analysis been performed appropriately and rigorously? 

Reviewer #1: Yes

Reviewer #2: Yes

3. Have the authors made all data underlying the findings in their manuscript fully available?

Reviewer #1: Yes

Reviewer #2: Yes

4. Is the manuscript presented in an intelligible fashion and written in standard English?

Reviewer #1: Yes

Reviewer #2: Yes

5. Review Comments to the Author

Reviewer #1: Dear Editor

The manuscript "High-resolution raindrop counting via instantaneous frequency sensing on hydrophobic elastic membranes" proposes a novel approach to improve the temporal resolution of raindrop counting, namely doing so by identifying impacts with high frequency transients in membranes made with elastic hydrophobic materials.

In particular, the paper contributes to two distinct aspects of the problem:

1. A handy acoustic model to detect instantaneous high frequency vibrations.

2. A machine learning framework that uses compact, physics-informed convolutional neural networks to improve the quality of detection.

The results are positive and interesting and has the potential to impact this field. The importance of such a project is clearly high, as it can serve as a basis for technological and further scientific discoveries.

One point I think is worth considering, and I believe would be a good idea to elaborate on the text is whether a fundamental understanding of the dynamical response of membranes might help such an effort. In particular when considering the work presented in

C. Steinbock, E. Katzav and A. Boudaoud, The Structure of Fluctuating Thin Sheets Under Random Forcing, Phys. Rev. Res. 4, 033096 (2022).

C. Steinbock and E. Katzav, The Dynamics of Fluctuating Thin Sheets Under Random Forcing, Phys. Rev. E 107, 025002 (2023).

Would such a principled calculation of noise in membranes enable further denoising?

I believe that an analysis of the points in this algorithm where a more fundamental approach could improve the results would greatly improve the interest and applicability of the paper.

I am happy to recommend publication of the paper once the author consider this.

Reviewer #2: The manuscript introduces a novel method for high-resolution raindrop counting using instantaneous frequency sensing on hydrophobic elastic membranes. This approach leverages microphone data processed by a compact machine-learning model to achieve a temporal resolution of 10 milliseconds with 80% accuracy. The proposed system aims to provide an affordable, high-precision rainfall sensor that overcomes the limitations of traditional rain sensing methods, such as the need for extensive power infrastructure and specialized expertise. This is an interesting work. I recommend the manuscript for publication after minor revisions to address the points below:

1. The manuscript is generally well-written, with clear and concise language. However, minor grammatical and typographical errors should be addressed.

2. The introduction section could be slightly condensed to improve readability and focus on the key contributions of the study.

3. Providing supplementary information on the experimental setup, including detailed specifications of the hardware components used, would be beneficial for readers interested in replicating the study.

4. The thickness, modulus, and some viscoelastic properties of the membrane should play an important role in sensing. The authors may try a few other geometries and properties to justify the accuracy and university of their methodology.

6. PLOS authors have the option to publish the peer review history of their article (what does this mean?). If published, this will include your full peer review and any attached files.

Reviewer #1: No

Reviewer #2: No

---

## [Author Response · Author response to Decision Letter 0]

23 Sep 2024

Dear Editor,

Thank you for a swift reviewing process.

I made my best attempt to correct the PLOS ONE style discrepancies according to the provided guidelines. 

Several references and one subsection have been added in response to Reviewers' comments.

No retracted or unpublished references are present. 

The acknowledgments section has been added too.

Please find a revised version of the manuscript and my responses to the Reviewers' comments.

I am grateful to both Reviewers for their feedback which undoubtedly led to improvements.

* Response to Reviewer #1

I am grateful for pointing to interesting physics of crumpling thin sheets (Mylar happens to be one of the two materials tested in my work). His/her question led to an additional reflection, which is summarized in the manuscript. 

Based on my knowledge of Foppl-von Karman equation, it is designed to address large deflections on thin elastic sheets such as 'crumpling' experiments, described in Lahini et al., PRL, 2023. In the second cited article, the response to forcing by Gaussian noise is analyzed. The cited articles build on a material and forcing models that do not describe the rain drop counter well, but it is nevertheless of interest to find possible parallels between the two contexts.

As a general response to "whether a fundamental understanding of the dynamical response of membranes might help such an effort" (of de-noising) is Yes. I believe that synergy of deep learning and fundamental understanding of physics could achieve better results with respect to brute force deep learning. This knowledge is practiced implicitly, through the choice of ML architectures, feature design, initializers, and loss functions that incorporate the properties of a particular two-point function and other known physical fact into the model. I would specifically like to point out the paradigm of 'algorithm unrolling' which has been well tested in image denoising but is relatively little known in acoustics. I also should remark that dealing with noise in practical applications is a delicate matter, because any type of environmental sound may, in a given setting, become (an unwanted) noise. A simple CNN approach might be too simplistic after all, and more sophisticated machinery, such as autoencoders, might be a better fit. Additional considerations based on the type of autocorrelation function could be: using dilated convolutions, using lagged loss functions or loss functions with power law constraints, using autoencoder or RNN based architectures, if the autocorrelation has long memory.

* Response to Reviewer #2

1. Typos, grammatical errors: Several grammatical errors and style improvements have been addressed.

2. Shorter introduction, describe the contributions: The introductory paragraphs have been trimmed. The contribution is thoroughly described on lines 31--55, and further context is provided in the remainder of the introduction. I would be happy to consider additional suggestions on how to make the text more accessible.

3. Hardware and software: a paragraph has been added at the end of Sect 2. I believe this issue is now fully addressed taking into account the supplementary materials provided in the accompanying GiHub repository, where python code and data used for this project could be found. YouTube videos on how to build a device and identify events are also available.

4. Missing material parameters and experimentation with different geometries: The requested material parameters are provided in Sect. 2 (first paragraph) and it is specified where a value (surface tension) could not be provided. We did try several geometries without any interesting new results. The geometry parameters affect the resonances but the main point of our methodology is that the important resonances are separated by a large frequency band/temporal scale, and therefore easily separable. Our conclusions are: 1) the effect on which the counting is based was observed in all these cases (thin, thick membrane, various surface tension, diameter values) so long as soft hydrophobic membranes are used, and 2) the details of the resonances induced by the aforementioned properties are mostly irrelevant to our methodology due to the time scale separation. The interesting question of finding the limits of applicability or the optimal parameters is not addressed, owing to limited access to specialized lab equipment.

---

## [Decision Letter · Decision Letter 1]

30 Sep 2024

High-resolution raindrop counting via instantaneous frequency sensing on hydrophobic elastic membranes

PONE-D-24-29859R1

Dear Dr. Paškauskas,

We’re pleased to inform you that your manuscript has been judged scientifically suitable for publication and will be formally accepted for publication once it meets all outstanding technical requirements.

Kind regards,

Zaky A. Zaky, Lecturer

Academic Editor

PLOS ONE

Additional Editor Comments (optional):

Reviewers' comments:

Reviewer's Responses to Questions

**Comments to the Author**

1. If the authors have adequately addressed your comments raised in a previous round of review and you feel that this manuscript is now acceptable for publication, you may indicate that here to bypass the “Comments to the Author” section, enter your conflict of interest statement in the “Confidential to Editor” section, and submit your "Accept" recommendation.

Reviewer #1: All comments have been addressed

Reviewer #2: All comments have been addressed

2. Is the manuscript technically sound, and do the data support the conclusions?

Reviewer #1: Yes

Reviewer #2: Yes

3. Has the statistical analysis been performed appropriately and rigorously? 

Reviewer #1: Yes

Reviewer #2: Yes

4. Have the authors made all data underlying the findings in their manuscript fully available?

Reviewer #1: Yes

Reviewer #2: Yes

5. Is the manuscript presented in an intelligible fashion and written in standard English?

Reviewer #1: Yes

Reviewer #2: Yes

6. Review Comments to the Author

Reviewer #1: Dear Editor

The author have addressed all the comments raised by both referees and I theefore recommend accepting it to PLOS ONE.

Reviewer #2: (No Response)

7. PLOS authors have the option to publish the peer review history of their article (what does this mean?). If published, this will include your full peer review and any attached files.

Reviewer #1: No

Reviewer #2: **Yes: **Zhaohe Dai

---

## [Editor Report · Acceptance letter]

3 Oct 2024

PONE-D-24-29859R1 

PLOS ONE

Dear Dr. Paškauskas, 

I'm pleased to inform you that your manuscript has been deemed suitable for publication in PLOS ONE. Congratulations! Your manuscript is now being handed over to our production team.

Kind regards, 

on behalf of

Dr. Zaky A. Zaky 

Academic Editor

PLOS ONE